# The “Jekyll Side” of the S100B Protein: Its Trophic Action in the Diet

**DOI:** 10.3390/nu17050881

**Published:** 2025-02-28

**Authors:** Fabrizio Michetti, Vincenzo Romano Spica

**Affiliations:** Department of Movement, Human, and Health Sciences, University of Rome “Foro Italico”, 00135 Rome, Italy; fabriziomichetti.office@gmail.com

**Keywords:** S100B protein, glial cells, neurotrophic factor, proinflammatory role, microbiota interaction, breast milk, trophic nutrient

## Abstract

The calcium-binding S100B protein is concentrated in glial cells (including enteroglial cells) in the nervous system. Its conformation and amino acid composition are significantly conserved in different species; this characteristic suggests conserved biological role(s) for the protein. The biological activity is concentration-dependent: low physiological concentrations exert a neurotrophic effect, while high concentrations exert a proinflammatory/toxic role. The proinflammatory/toxic role of S100B currently attracts the scientific community’s primary attention, while the protein’s physiological action remains unraveled—yet remarkably interesting. This is now a topical issue due to the recently consolidated notion that S100B is a natural trophic nutrient available in breast milk and/or other aliments, possibly interacting with other body districts through its impact on microbiota. These recent data may offer novel clues to understanding the role of this challenging protein.

## 1. Introduction

The S100B protein is a calcium-binding protein concentrated in glial cells (mainly astrocytes) in the nervous system. The protein is also characterized by its solubility in a 100% saturated solution with ammonium sulfate. This peculiarity was at the basis of its denomination, which was originally the S100 protein. Interestingly, the conformation and amino acid composition of S100B have been shown to be significantly conserved in different species, suggesting that it may have a crucially conserved biological role and, possibly, a functional specificity [1]. This protein is part of the S100 protein family, which has over 20 calcium-binding proteins. They exhibit structural similarities and modulate the activity of a large number of targets operating as calcium-activated switches in different tissues. S100 proteins constitute a subgroup within the EF-hand protein superfamily, which is characterized by a calcium-binding loop forming a conserved pentagonal arrangement around the calcium ion (the EF-hand motif). Each S100 protein is characterized by the presence of two Ca^2+^-modulated motifs of the EF-hand type interconnected by an intermediate region, which is often referred to as the hinge region, resulting in a helix–loop–helix arrangement. S100B is an acidic homodimer (2 beta subunits) of 9–14 kDa per monomer that constituted the bulk of the protein fraction originally isolated from brain extracts in the early 1960s. Using chromatographic and electrophoretic methods, the protein originally was detectable in brain extracts but not in non-neural tissues; consequently, it was regarded as specific to the nervous system. In the nervous system, S100B has been shown to be concentrated in astrocytes, as above indicated, and other glial cell types, namely enteric glial cells, and it has also been reportedly located in some neuron subpopulations. The protein was later identified in definite extra-neural cell types, such as chondrocytes, melanocytes, Langerhans cells, dendritic cells of lymphoid organs, some lymphocyte cell types, adrenal medulla satellite cells, skeletal muscle satellite cells, tubular kidney cells, Leydig cells, non-nervous structures of the eye, and adipocytes. Additionally, S100B-like immunoreactivity has been detected in invertebrates, such as the planarian (*Dugesia gonocephala*), the silkworm (*Bombyx mori*), the earthworm (*Pheretima communissima*), the cockroach (*Periplaneta americana*) [2,3], and, most importantly, in spinach (*Spinacia oleracea*) and other plants [4,5]. These latter findings, together with the detection of S100B in milk [6,7], are of particularly relevant interest considering the novel view indicating that this protein is a natural component of different aliments, with a potential role as a dietary driver.

To date, the activity of S100B has been studied in different tissues and biological fluids. It has appeared to be closely associated with its concentration. In particular, low (nanomolar) concentrations of the protein, which are regarded as physiologic and were the original focus of research, have been found to exert a trophic effect (the “Jekyll side” of S100B), while high (micromolar) concentrations of the protein appear to exert a toxic/proinflammatory role (the “Hyde side” of S100B), behaving as a danger/damage-associated molecular pattern (DAMP) protein [1,8,9]. In either case, extracellular S100B is regarded as interacting with surrounding cell types mainly through the Receptor for Advanced Glication Endproducts (RAGE), a ubiquitous, transmembrane, immunoglobulin-like receptor that binds to a diverse range of extracellular ligands and intracellular effectors [10]. At present, the “Hyde side” of S100B appears to attract the primary interest of the scientific community, especially regarding the use of the protein levels as a biomarker of different disorders, such as Alzheimer’s disease (global prevalence of dementia: 0.4‰) and multiple sclerosis (global prevalence: 0.04‰), but also as a pathogenic factor and, as a consequence, as a putative therapeutic target. Among other possibilities, some members of the S100 protein family (S100A9) have been attributed roles in neurogenerative diseases due to their fibrillization under the effect of calcium divalent ions [11]. At present, S100B is also regarded as a biomarker in assessing a tumor’s load, stage, and prognosis [12,13].

This interest, which undeniably deserves to be intensely cultivated, has originated from the direct application of the related findings to pathological conditions, which may include extreme or urgent clinical conditions. However, the physiological role of this protein (the “Jekyll side”), which likely constitutes the main biological (trophic) function of S100B and is far from being understood in depth, may offer information that could form the foundation for new insights and potential applications. The physiological role of the protein was intensively investigated soon after its discovery, and it is now particularly salient in light of the recent discovery that the protein is also present in foods and represents a nutritional driver that could display a trophic effect. In fact, the protein is present in milk from different mammalian species and in widely diffused aliments such as dairy products, vegetables, and fruits [5,6]. In this respect, an effect on the biodiversity of the gut microbiota [14] has been shown [15,16]. This Perspective paper intends to draw attention to the “Jekyll side” of S100B, emphasizing its presence in the diet, and stimulate research in this novel direction.

## 2. The S100B Protein as a Protective and Trophic Factor

Many studies that are reported in this section date back several years, but they deserve now to be reconsidered. The earlier observational findings, published soon after the discovery of the protein, identified S100B (at that time known as S100) as a molecule playing a key, possibly trophic, role in biological organisms. In particular, the protein was shown to conserve its immunological features during phylogenesis, as indicated above [17,18,19]. Interestingly, during ontogenesis, the pattern of its accumulation was shown to parallel the functional maturation of regions of the central nervous system in different species, including humans. In particular, S100B levels appeared to be elevated during the period of glial cell proliferation and neuronal differentiation [20,21,22,23,24,25]. More recently, S100B regulation has been reported to affect the differentiation during neurodevelopment [26]. Several mechanistic studies, in many cases in vitro, were then performed in order to investigate the processes accompanying the physiological/trophic role proposed for the S100B protein (Figure 1). Thus, S100B was shown to stimulate neurite outgrowth [27,28,29] and enhance cell survival [29,30] in cultures of neurons and glial cells. The extracellular trophic activities of S100B were considered to require a disulfide-linked, dimeric form of the protein [31], while the antisense inhibition of glial production of the S100B constitutive monomer (S100b) resulted in alterations in cell morphology, cytoskeletal organization, and cell proliferation [32]. A series of papers, distributed over approximately two decades, proposed a role for the nanomolar concentrations (regarded as physiological) of S100B in adult neurogenesis [33]. An intraventricular S100B infusion (50 ng) was shown to induce neurogenesis within the hippocampus, which could be associated with enhanced cognitive functioning, following an experimental traumatic brain injury (lateral fluid percussion) in rats [34]. In mice subjected to an experimental brain injury (unilateral parietal cryolesion), the intraperitoneal treatment with a nanomolar concentration of S100B significantly enhanced the early progenitor cell proliferation in the hippocampal subgranular zone (SGZ), the germinal area of the hippocampus, as well as the cell survival and migration to the granule cell layer (GCL) and promoted neuronal differentiation [35]. Likewise, the treatment with a nanomolar concentration of S100B for 7 days using osmotic micropumps augmented the hippocampal neurogenesis and synaptogenesis in rats after a traumatic brain injury (lateral fluid percussion) [36]. Moreover, long-term increased nanomolar S100B levels have also been shown to promote progenitor cell proliferation in the SGZ in juvenile, adult, and one-year-old S100B transgenic mice, as well as migration to the GCL in older S100B transgenic mice, suggesting a neurotrophic role in neuroplasticity, neurogenesis, and neuroregenerative medicine [37]. A protective role for the protein has also been proposed in Alzheimer’s disease (AD) alongside findings that indicate its pathogenic role at high concentrations and its role as a biomarker in biological fluids [1]. Thus, S100B has been shown to suppress the Aβ42 and Tau aggregation and toxicity by acting as a chaperone and metal ion buffering protein [38,39,40,41,42]. S100B was also shown to reduce the levels of inflammatory cytokines, such as Il-17 and IFN-a2, overexpressed after treatment with Aβ42, in cultured astrocytes, suggesting that the susceptibility of S100B to oxidation influences its protective activities [43]. Despite the reported findings that support a physiological role for S100B, a univocal function has not yet been established. Presently, the extracellular trophic activities have generated increased interest in light of the novel view interpreting S100B as a nutritional driver.

## 3. The S100B Protein as a Nutrient in Different Aliments

### 3.1. Milk

A relevant finding that supports a trophic role for S100B involves its detection in human breast milk [6,7] since the nutrient composition of human milk is considered to be essential for the growth, development, and health of infants, and breastfeeding for newborns is recognized as providing the infant with an optimal start in life [44,45]. Thus, S100B, as a putative trophic factor, might be considered among the mechanisms responsible for the beneficial effects of breastfeeding [46]. The natural role of S100B in milk appears not to be restricted to humans, since the protein has been identified in the milk produced by other mammalian species, namely donkeys, sheep, goats, and cows, although at a lower concentration than in humans [47]. It should be noted that S100B was shown to be essentially lost in milk formulae and donor milk subjected to Holder pasteurization, likely due to the preparation procedures [48,49].

### 3.2. Edible Plants

In addition, after an earlier study indicating the immunochemical and immunohistochemical detection of S100B-like immunoreactivity in *Spinacia oleracea* tissues [4], this protein was identified in diffused edible plants both in silico and in vivo. In particular, the S100B motif was expected in silico in the genome of some species belonging to the *Viridiplantae* clade, including various widely diffused edible plants such as *Malus domestica*, *Durio zibethinus*, and *Artocarpus heterophyllus*. S100B-like immunoreactive material was also detected in samples from fruits (e.g., apples, durian, jackfruit, and their derivatives) or leaves of the same species [5]. These recent findings corroborate the concept that S100B is naturally present in several foods. As a consequence, the possibility that S100B naturally exerts a nutritional and trophic function in the diet appears reasonable (Table 1) [50].

### 3.3. Gut Microbiota Interactions

The question remains how the S100B protein, when consumed with milk or other aliments, may interact with the tissues of the host that assumed it, since data regarding its gut absorption are not available. A possible explanation comes from recent findings indicating a novel role for the protein within the intestinal lumen, as S100B has been shown to affect gut microbiota biodiversity. Gut microbiota are known to consist of millions of microorganisms present in the human intestinal apparatus, playing a key role in food digestion, immune control, and antitumor responses, as well as in the synthesis of beneficial compounds and the control of neural regulation [14]. Based on the microbiota composition, proteins putatively interacting with S100B domains were identified in silico, both in healthy subjects and in patients with inflammatory bowel disease. Interestingly, differences in the interacting domain occurrences were exhibited between the two groups [15]. These results led to the experimental investigation of the role of S100B as a candidate signaling molecule in the gut microbiota communication machinery. Thus, the in silico data were experimentally confirmed in mice, in which the S100B levels correlated with the microbiota biodiversity. The correlation was significantly reduced after treatment with the S100B inhibitor pentamidine [51], indicating that the correlation was likely influenced by the modulation of S100B activity [16]. Taken together, these data furnish the missing link in the chain: microbiota-dependent processes might mediate the S100B influences on tissues of the host, including the nervous system through the gut–brain axis (Figure 2).

It should also be noted that, as an additional or alternative source of S100B for interactions with microbiota, S100B is known to be a constituent of enteroglial cells, corresponding to astrocytes in the enteric nervous system [52]. Once released by the enteroglial cells (S100B is known to be released by astrocytes in the central nervous system), S100B might trigger phenomena that are dependent on microbiota activation. In this way, as a hypothesis, an S100B-dependent influence mediated by microbiota might occur in tissues even when the protein is not ingested through aliments.

## 4. Conclusions

The novel perspective interpreting the S100B protein as a nutrient possibly displaying trophic effects valorizes the physiological role of the protein. This side of the protein (the “Jekyll side”) has been underestimated in recent times, probably since urgent needs based on pathological conditions have been more compelling (the “Hyde side”). In light of the presence of S100B in aliments, one might even consider the possibility that S100B in nature plays a role as a nutritional driver in living beings, including humans, from the neonatal stage, when it is ingested through milk, through the adult stages if a healthy diet rich in vegetables, fruits, and dairy products is followed. The interaction of the protein with gut microbiota offers a possible explanation for the role of enteric S100B, ingested with aliments and/or secreted from enteroglial cells. This novel perspective may constitute a fruitful line of research, potentially opening unexpected views of the physiological role(s) of this challenging protein.

## Figures and Tables

**Figure 1 nutrients-17-00881-f001:**
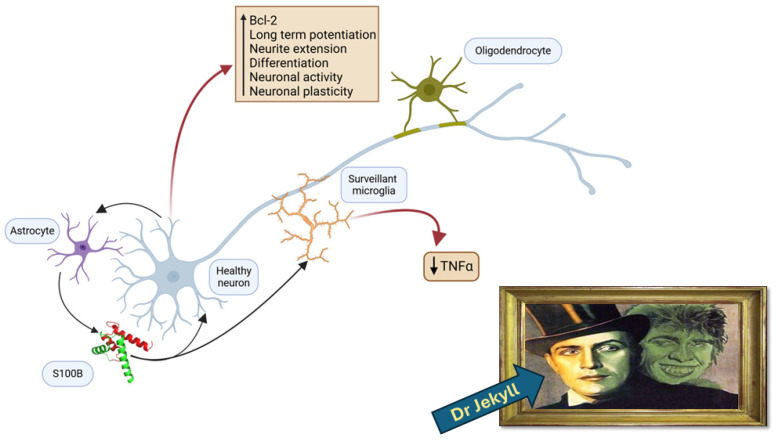
The “Jekyll side” of S100B. The trophic function of S100B in the nervous system involves astrocytes as the main source of the protein, oligodendrocytes for myelination processes, microglia for surveillance processes and TNFα regulation, and neurons as the final targets. The functional pathways include the antiapoptotic Bcl-2 increase, long-term potentiation, neurite extension, stimulation of neuronal activity, plasticity, and differentiation. (Created with BioRender.com).

**Figure 2 nutrients-17-00881-f002:**
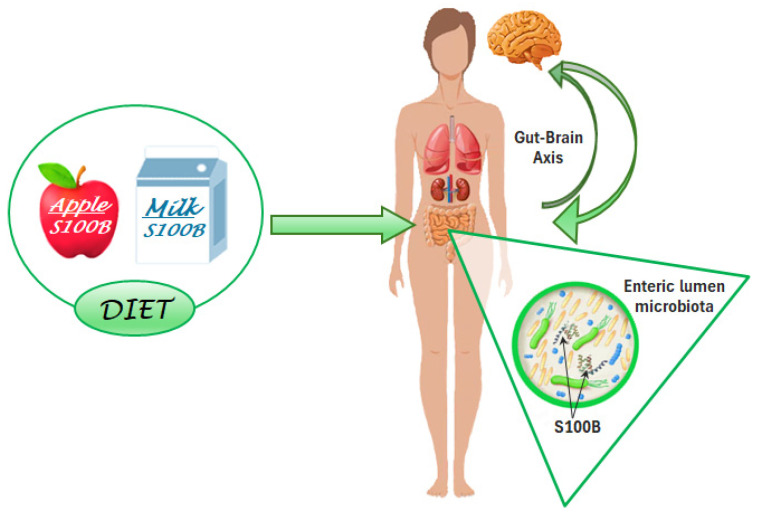
S100B is present in the diet and exerts positive effects. Common aliments, including apples and milk, as well as various edible plants and dairy products, contain S100B. The positive Jekyll effects are mediated by an action on gut microbiota biodiversity. When ingested, S100B may influence the gut–brain axis—and possibly other axes regulating different body districts. (Created with BioRender.com).

**Table 1 nutrients-17-00881-t001:** Concentration of S100B protein in various food sources.

Food Source	S100B Concentration
Farm animal milk	0.03–180 µg/L
Cheese–Dairy	0.01–0.4 µg/Kg
Fruit–Vegetables	0.04–180 µg/Kg

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
