# Peer review of "The “Jekyll Side” of the S100B Protein: Its Trophic Action in the Diet"

_nutrients, 2025, doi:10.3390/nu17050881_

Round 1
Reviewer 1 Report
Comments and Suggestions for Authors
The manuscript titled “The “Jekyll side” of S100B protein: the trophic actiion in diet” by Michetti, F.; et al. is a scientific work where the authors provide some perspectives in the field of S100B protein family and their exerted action as dietary nutrient which is in contrast to its proinflammatory and neurodegenerative action. The manuscript is generally well-written and this is a topic of growing interest.
However, it exists some points that need to be addressed (please, see them below detailed point-by-point) to improve the scientific quality of the submitted manuscript paper before this article will be consider for its publication in Nutrients.
1) Keywords. The authors should add some relevant terms according to this perspective work in the keyword list.
2) “This protein is known to be a member of the S100 protein family (…) bulk of the protein fraction originally isolated from brain extracts in the early ‘60s” (lines 24-35). Here, it would be also neccesary how the S100 protein family can lead to fibrillization that causes neurodegenerative diseases under the effect of calcium divalent ions [1]. Furthermore, it was recently reported that the screening of this protein family can be exploited as marker of certain cancer malignancies [2]
[1] https://doi.org/10.3390/biom14091091
[2] https://doi.org/10.3390/biom13030529
3) “At the present, the “Hyde side” of S100B appears (…) biomarker of different disorders, but also, as pathogenic factor, and, as a consequence, as a putative therapeutic targe” (lines 63-66). What pathogenic disorders? Could the authors provide quantitative data insights about the worldwide global burdens of incidence of these disorders? This information could serve the potential readers to better understand the significance of this work.
4) “2. The S100B as a protective and trophic factor” (lines 83-131). A schematic representation highlighted the underlying molecular mechanisms of S100B to exert their trophic function will benefit the potential readers.
5) “3. The S100B as a nutrient in different aliments” (lines 132-195). Some quantification about the S100B availability concentrations in different food resources is required. For it, a summarizing table will clearly illustrate the expected S100B amount thresholds in different aliments.
6) “4. Conclusions” (lines 196-207). This section perfectly remarks the most relevant outcomes found by the authors in this work and also the promising future prospectives. It may be advisable to add a brief statement to remark the potential future action lines to pursue the topic covered in this research.
Author Response
Dear Editor,
We would like to thank you and both reviewers for kind comments and useful indications, which have been taken into due account in the revised text.
Reviewer 1:
Comments and Suggestions for Authors
The manuscript titled “The “Jekyll side” of S100B protein: the trophic actiion in diet” by Michetti, F.; et al. is a scientific work where the authors provide some perspectives in the field of S100B protein family and their exerted action as dietary nutrient which is in contrast to its proinflammatory and neurodegenerative action. The manuscript is generally well-written and this is a topic of growing interest.
However, it exists some points that need to be addressed (please, see them below detailed point-by-point) to improve the scientific quality of the submitted manuscript paper before this article will be consider for its publication in Nutrients.
1) Keywords. The authors should add some relevant terms according to this perspective work in the keyword list.
REPLAY: Keywords have been added to the text, according to the indications suggested by Reviewer 2.
2) “This protein is known to be a member of the S100 protein family (…) bulk of the protein fraction originally isolated from brain extracts in the early ‘60s” (lines 24-35). Here, it would be also neccesary how the S100 protein family can lead to fibrillization that causes neurodegenerative diseases under the effect of calcium divalent ions [1]. Furthermore, it was recently reported that the screening of this protein family can be exploited as marker of certain cancer malignancies [2]
[1] https://doi.org/10.3390/biom14091091
[2] https://doi.org/10.3390/biom13030529
REPLAY: the indications suggested by the referee have been added to the text, thus also improving the introduction section.
3) “At the present, the “Hyde side” of S100B appears (…) biomarker of different disorders, but also, as pathogenic factor, and, as a consequence, as a putative therapeutic targe” (lines 63-66). What pathogenic disorders? Could the authors provide quantitative data insights about the worldwide global burdens of incidence of these disorders? This information could serve the potential readers to better understand the significance of this work.
REPLAY: the indications suggested by the referee have been added to the text, thus also improving the introduction section.
4) “2. The S100B as a protective and trophic factor” (lines 83-131). A schematic representation highlighted the underlying molecular mechanisms of S100B to exert their trophic function will benefit the potential readers.
REPLAY: Figure 1 has been added, according to the request of the reviewer.
5) “3. The S100B as a nutrient in different aliments” (lines 132-195). Some quantification about the S100B availability concentrations in different food resources is required. For it, a summarizing table will clearly illustrate the expected S100B amount thresholds in different aliments.
REPLAY: Table 1 has been added according to the request of the reviewer.
6) “4. Conclusions” (lines 196-207).This section perfectly remarks the most relevant outcomes found by the authors in this work and also the promising future prospectives. It may be advisable to add a brief statement to remark the potential future action lines to pursue the topic covered in this research.
REPLAY: Thank you for your comment, a brief statement has been added according to reviewer’s suggestion.
Reviewer 2 Report
Comments and Suggestions for Authors
The authors of this manuscript aim to discuss the dual role of the S100B, emphasizing its lesser-studied physiological function rather than its well-documented pathological effects. S100B is a calcium-binding protein primarily found in glial cells of the nervous system but also present in various other cell types and even in non-neural tissues. Its structure and amino acid sequence are highly conserved across species, suggesting an essential biological role. The protein exhibits a "Jekyll and Hyde" duality: at low (nanomolar) concentrations, it exerts a neurotrophic and protective effect and at high (micromolar) concentrations, it acts as a proinflammatory molecule, behaving as a damage-associated molecular pattern protein via its interaction with the Receptor for Advanced Glycation Endproducts. This paper seeks to shift attention toward the beneficial "Jekyll side" of S100B, particularly its role in nutrition and microbiota interactions, encouraging further research into its potential as a dietary factor with biological significance.
The manuscript would benefit from further improvement by offering a more in-depth discussion of the data presented. Currently, the analysis appears to be somewhat superficial, and a more comprehensive exploration of the recent studies would provide an expanded perspective for the reader. A deeper examination of the data could help to better illustrate the underlying mechanisms and potential implications, as well as discuss any limitations. Expanding discussion would not only strengthen the manuscript but also allow a better understanding of the importance of the data presented.
I have a few suggestions regarding the manuscript, detailed as follows:
· Please ensure that citations are formatted correctly according to the guidelines provided in the author's instructions!
· Please ensure that all the citations are up-to-date and reference the most recent relevant research.
· Consider adding a table summarizing the data on the effects of the S100B protein in diet to enhance comprehension.
· Please include at least one additional figure that provides a molecular explanation.
· Ensure consistent use of terms like "S100B protein" and "S100B" throughout the manuscript. For example, "S100B protein" could simply be referred to as "S100B" after the initial introduction.
· Please expand the section about the physiological role of S100B.
· The manuscript should summarize the key findings and discuss their broader implications more explicitly. It currently ends with a broad statement about the novel role of S100B but does not provide a clear synthesis of the findings or suggestions for future research.
· The manuscript seems to lack a clear structure in some areas. It moves between different ideas, such as S100B’s role in milk, its presence in various plants, and its relationship with gut microbiota, without smooth transitions. It would be helpful to break these sections into more distinct paragraphs, each focused on one concept.
· Line 14-15: Please add keywords to the abstract to improve its searchability and relevance for readers and researchers. Consider including terms such as S100B protein, glial cells, neurotrophic factor, proinflammatory role, microbiota interaction, breast milk, and trophic nutrient.
· Line 211: Remove “Please add:”
· Line 215-217: Please ensure you include the appropriate text in this section: Acknowledgments.
· Line 31: Please change Ca2+ to Ca2+
· Line 206-207: Please consider removing citation from the conclusion.
· A more detailed discussion on the effects of the S100B protein in diet should be included to strengthen the analysis.
· Line 222: If the Appendix is not being used, please consider removing it to streamline the document.
Comments on the Quality of English LanguageThe quality of the English Language in the manuscript could be improved to ensure better understanding of the data presented.
Author Response
Reviewer 2:
Comments and Suggestions for Authors
The authors of this manuscript aim to discuss the dual role of the S100B, emphasizing its lesser-studied physiological function rather than its well-documented pathological effects. S100B is a calcium-binding protein primarily found in glial cells of the nervous system but also present in various other cell types and even in non-neural tissues. Its structure and amino acid sequence are highly conserved across species, suggesting an essential biological role. The protein exhibits a "Jekyll and Hyde" duality: at low (nanomolar) concentrations, it exerts a neurotrophic and protective effect and at high (micromolar) concentrations, it acts as a proinflammatory molecule, behaving as a damage-associated molecular pattern protein via its interaction with the Receptor for Advanced Glycation Endproducts. This paper seeks to shift attention toward the beneficial "Jekyll side" of S100B, particularly its role in nutrition and microbiota interactions, encouraging further research into its potential as a dietary factor with biological significance.
The manuscript would benefit from further improvement by offering a more in-depth discussion of the data presented. Currently, the analysis appears to be somewhat superficial, and a more comprehensive exploration of the recent studies would provide an expanded perspective for the reader. A deeper examination of the data could help to better illustrate the underlying mechanisms and potential implications, as well as discuss any limitations. Expanding discussion would not only strengthen the manuscript but also allow a better understanding of the importance of the data presented.
I have a few suggestions regarding the manuscript, detailed as follows:
- Please ensure that citations are formatted correctly according to the guidelines provided in the author's instructions!
- Please ensure that all the citations are up-to-date and reference the most recent relevant research.
REPLAY: in the present updated version, citations were corrected and formatted accordingly.
- Consider adding a table summarizing the data on the effects of the S100B protein in diet to enhance comprehension.
REPLAY: in the present version a figure has been added concerning the trophic mechanisms of action of the protein and also a table concerning protein concentration in different aliments; an additional figure has been proposed also as graphical abstract summarizing the putative mechanisms/effects.
- Please include at least one additional figure that provides a molecular explanation.
REPLAY: in the present version a figure has been added concerning the trophic mechanisms of action of the protein, as previously indicated.
- Ensure consistent use of terms like "S100B protein" and "S100B" throughout the manuscript. For example, "S100B protein" could simply be referred to as "S100B" after the initial introduction.
REPLAY: Thank you for this observation. It was checked and revised accordingly.
- Please expand the section about the physiological role of S100B.
REPLAY: the text was updated accordingly.
- The manuscript should summarize the key findings and discuss their broader implications more explicitly. It currently ends with a broad statement about the novel role of S100B but does not provide a clear synthesis of the findings or suggestions for future research.
REPLAY: it was revised and modified accordingly.
- The manuscript seems to lack a clear structure in some areas. It moves between different ideas, such as S100B’s role in milk, its presence in various plants, and its relationship with gut microbiota, without smooth transitions. It would be helpful to break these sections into more distinct paragraphs, each focused on one concept.
REPLAY: additional paragraphs were added and text updated as suggested.
- Line 14-15:Please add keywords to the abstract to improve its searchability and relevance for readers and researchers. Consider including terms such as S100B protein, glial cells, neurotrophic factor, proinflammatory role, microbiota interaction, breast milk, and trophic nutrient.
REPLAY: thanks a lot for the observation and for the helpful suggestion.
- Line 211: Remove “Please add:”
REPLAY: done
- Line 215-217:Please ensure you include the appropriate text in this section: Acknowledgments.
REPLAY: THANKS A LOT! done
- Line 31:Please change Ca2+ to Ca2+
REPLAY: done
- Line 206-207: Please consider removing citation from the conclusion.
REPLAY: done
- A more detailed discussion on the effects of the S100B protein in diet should be included to strengthen the analysis.
REPLAY: done
- Line 222:If the Appendix is not being used, please consider removing it to streamline the document.
REPLAY: done
The quality of the English Language in the manuscript could be improved to ensure better understanding of the data presented.
REPLAY: done
Round 2
Reviewer 1 Report
Comments and Suggestions for Authors
The authors did a great deal of effort to cover all the suggestions raised by the Reviewers. For this reason, the scientific manuscript quality was greatly improved. Based on the novelty and significance of the gathered results, this work can be published in Nutrients.
Author Response
Thank you, for your helpful observations that allowed a satisfying improvement of the manuscript.
Reviewer 2 Report
Comments and Suggestions for Authors
1. The manuscript would significantly benefit from a more in-depth discussion of the presented data and relevant recent studies. Currently, the discussion is superficial and expanding it would strengthen the overall quality of the manuscript while improving the reader’s understanding of the significance of the data presented.
2. Additionally, a thorough English revision is required, as there are numerous grammatical and linguistic errors that affect clarity and readability. Particular attention should be given to correcting typos such as “ailments” and ensuring consistent and accurate language throughout the text.
3. The abstract requires improvement to enhance clarity, it should provide a more comprehensive summary of the study, including implications. Additionally, the language and structure should be refined for better readability.
4. Regarding the figures, Figure 2 needs improvement, and any existing typos should be corrected. For Figure 1, unnecessary spaces should be eliminated, and the application used to create the figures should be clearly stated. The same applies to Figure 2.
5. Citations have not been properly formatted in accordance with the author’s guidelines. Ensure that all references are correctly cited and adhere to the required formatting style.
6. The discussion section requires further improvement to enhance the depth and critical analysis of the findings. A more thorough evaluation of the data, along with a discussion of its implications is necessary.
7. The table included in the manuscript appears too basic and should be expanded to provide more information.
8. Please ensure that the proper numbering system is used for paragraphs to maintain a clear and organized structure throughout the manuscript.
Comments on the Quality of English LanguageThe English language needs improvement, all typographical errors should be corrected. Please pay attention to grammar, numbering and punctuation throughout the manuscript.
Author Response
- The manuscript would significantly benefit from a more in-depth discussion of the presented data and relevant recent studies. Currently, the discussion is superficial and expanding it would strengthen the overall quality of the manuscript while improving the reader’s understanding of the significance of the data presented.
R: In the present revised form of the manuscript we have essentially rewritten chapter 3a,b,c, which deals with more novel data and interpretations on the role of S100B as a nutrient in different aliments, which constitutes the bulk of the manuscript . We trust that the present form improves in fact the reader’s understanding of the significance of the data presented.
- Additionally, a thorough English revision is required, as there are numerous grammatical and linguistic errors that affect clarity and readability. Particular attention should be given to correcting typos such as “ailments” and ensuring consistent and accurate language throughout the text.
R: Done
- The abstract requires improvement to enhance clarity, it should provide a more comprehensive summary of the study, including implications. Additionally, the language and structure should be refined for better readability.
R: In the present revised form of the manuscript we ameliorated the clarity of the manuscript, providing a more comprehensive summary of the study, including implications.
- Regarding the figures, Figure 2 needs improvement, and any existing typos should be corrected. For Figure 1, unnecessary spaces should be eliminated, and the application used to create the figures should be clearly stated. The same applies to Figure 2.
R: Thanks, done.
- Citations have not been properly formatted in accordance with the author’s guidelines. Ensure that all references are correctly cited and adhere to the required formatting style.
R: Done.
- The discussion section requires further improvement to enhance the depth and critical analysis of the findings. A more thorough evaluation of the data, along with a discussion of its implications is necessary.
R: We assume that the Reviewer refers to the Conclusions section, since a Discussion section is not included in this Perspective/Review article. In any case, we have essentially rewritten the Conclusions section. We trust that the present form improves in fact the reader’s understanding of the significance of the data presented.
- The table included in the manuscript appears too basic and should be expanded to provide more information.
R: Yes, we agree with the observation that the table is basic, but it respects data presently available.
- Please ensure that the proper numbering system is used for paragraphs to maintain a clear and organized structure throughout the manuscript.
R: Done
Comments on the Quality of English Language
The English language needs improvement, all typographical errors should be corrected. Please pay attention to grammar, numbering and punctuation throughout the manuscript.
R: Done